# Improving the Estimation of PM$_{2.5}$ Concentration in the North China Area by Introducing an Attention Mechanism into Random Forest

Luo Zhang [1,2] , Zhengqiang Li [1,2,3,*] , Jie Guang [1] , Yisong Xie [1], Zheng Shi [4], Haoran Gu [1,3] and Yang Zheng [1]

1   State Environmental Protection Key Laboratory of Satellite Remote Sensing, Aerospace Information Research Institute, Chinese Academy of Sciences, Beijing 100101, China; zhangluo@aircas.ac.cn (L.Z.)
2   Key Laboratory of Earth Observation of Hainan Province, Hainan Research Institute, Aerospace Information Research Institute, Chinese Academy of Sciences, Sanya 572029, China
3   University of Chinese Academy of Sciences, Beijing 100049, China
4   The Administrative Center for China's Agenda 21, Beijing 100038, China
*   Correspondence: lizq@radi.ac.cn

**Abstract:** Fine particulate matter with an aerodynamic diameter less than 2.5 μm (PM$_{2.5}$) profoundly affects environmental systems, human health and economic structures. Multi-source data and advanced machine or deep-learning methods have provided a new chance for estimating the PM$_{2.5}$ concentrations at a high spatiotemporal resolution. In this paper, the Random Forest (RF) algorithm was applied to estimate hourly PM$_{2.5}$ of the North China area (Beijing–Tianjin–Hebei, BTH) based on the next-generation geostationary meteorological satellite Himawari-8/AHI (Advanced Himawari Imager) aerosol optical depth (AOD) products. To improve the estimation of PM$_{2.5}$ concentration across large areas, we construct a method for co-weighting the environmental similarity and the geographical distances by using an attention mechanism so that it can efficiently characterize the influence of spatial–temporal information hidden in adjacent ground monitoring sites. In experiment results, the hourly PM$_{2.5}$ estimates are well correlated with ground measurements in BTH, with a coefficient of determination (R$^2$) of 0.887, a root-mean-square error (RMSE) of 18.31 μg/m$^3$, and a mean absolute error (MAE) of 11.17 μg/m$^3$, indicating good model performance. In addition, this paper makes a comprehensive analysis of the effectiveness of multi-source data in the estimation process, in this way, to simplify the model structure and improve the estimation efficiency of the model while ensuring its accuracy.

**Keywords:** PM$_{2.5}$; random forest;attention mechanism; spatiotemporal prediction; multi-source data





## 1. Introduction

China's economy has developed rapidly in the past few decades, but it also faced serious air pollution problems, especially in the Beijing–Tianjin–Hebei (BTH) region [1]. Fine particulate matter with aerodynamic diameters less than 2.5 μm (PM$_{2.5}$) has been a major issue of public concern [2], associated with the risk to public health [3] and the impact on climate change [4] and economic structures [5,6]. Three common methods, such as ground station measurements, model simulations and satellite estimations can all provide PM$_{2.5}$ data. However, differences in data quality make it difficult for a single data source to meet high-precision data requirements.

Currently, numerous studies have estimated the concentration of fine particulate matter by employing satellite-based aerosol optical depth (AOD) products, and the data quality is improved by integrating other multi-source data [7–9]. Moderate Resolution Imaging Spectroradiometer (MODIS) AOD products are the most widely available data for PM$_{2.5}$ estimation [10–12]. In the early studies, the Dark Target (DT) and Deep Blue (DB) aerosol retrieval algorithms in MODIS provided global daily AOD products only at 10-km.

The application of these products in studies regarding $PM_{2.5}$ is limited by their coarse spatial resolution, such as the first global mapping about $PM_{2.5}$ spatia-l-temporal distribution [13]. With the improvement of the AOD retrieval algorithm, the quality of AOD and $PM_{2.5}$ data has also been improved [14–17], especially the daily AOD at 1 km produced by the multi-angle implementation of atmospheric correction (MAIAC) algorithms [18–20]. In addition, some $PM_{2.5}$ products using geostationary satellite AOD data can reach hourly levels, such as Himawari-8/AHI (Advanced Himawari Imager) [21,22] and Fengyun-4A/AGRI (the Advanced Geosynchronous Radiation Imager) [23,24], but the spatial resolution is still limited.

In most AOD-$PM_{2.5}$ estimation models, the minimum data unit is a vector of the site $PM_{2.5}$ value and the corresponding one-dimensional correlation factor (e.g., AOD, meteorological). Therefore, both linear and nonlinear models are fitted to a class of tabular feature data [25]. Some statistical regression models are simple and easy to implement, such as linear regression (LR) [26], linear mixed-effects models (LME) [27], geographically weighted regression (GWR) [14], geographically and temporally weighted regression (GTWR) [28], and some hybrid models [29–31]. Essentially, these liner models with local variation have difficulty capturing complex spatiotemporal relationships and have limited accuracy. In recent years, some deep learning and data-driven methods achieved promising results in remote sensing [32–35]. Machine learning methods, with their powerful ability to establish complex nonlinear relationships between various interacting predictor variables, are emerging as the dominant estimation method, such as artificial neural networks [36] and geo-intelligent deep belief networks [37].

In particular, some decision-tree-based models are more advantageous in fitting data with tabular features, such as the random forest (RF) [11,19,21], the Light Gradient Boosting Machine (LightGBM) [22], and the eXtreme Gradient Boosting (XGBoost) [38]. They are based on the idea of bootstrap aggregating and the random subspace method [39]. It aggregates a set of weak learners to form a strong one, and these weak learners, i.e., basic decision trees, are trained by randomly bootstrapped samples from the training set. Moreover, the node splitting features are also randomly selected during the training process of the decision tree. Thus, it can nonparametrically model the indefinable and complex non-linear relationship among input features. However, these decision-tree-based models are difficult to capture spatio-temporal correlation features of neighboring sites. Therefore, it is an important way to extract the spatio-temporal correlation via other different methods, as well as to construct a hybrid model for $PM_{2.5}$ estimation. In addition, in contrast to some other deep learning methods in the study of intelligent processing of remote sensing information [40,41], the full exploitation of multi-source data as well as spatio-temporal information plays a key role in the efficiency of the model [42,43].

There are two main hybrid model ways to achieve better $PM_{2.5}$ concentration estimation [8]: one is to fuse multiple sources of data to improve spatio-temporal resolution, and the other is to assemble other methods for capturing spatio-temporal correlation features of neighboring sites. Typically, these data fusion methods simply complement the spatio-temporal missing data, and it is difficult to verify the effectiveness of the data. The most essential characteristic of $PM_{2.5}$ is its spatial and temporal heterogeneity. Many researchers have tried to solve this problem by using geographical association information, which is constructed by weighting spatio-temporal distance from ground-based $PM_{2.5}$ measurements [19,20,37]. However, ground-based $PM_{2.5}$ monitoring sites are usually sparsely distributed in urban areas. Common weight methods with spatio-temporal distance is prone to over fitting in training. Moreover, its verifiability and effectiveness will be dramatically reduced when actually applied to large scale $PM_{2.5}$ concentration estimation.

In this paper, we improve the estimation of $PM_{2.5}$ concentration across large areas using multi-source data by aggregating random forest and the attention mechanism. The attention mechanism-based method is proposed to characterize the influence of spatial–temporal information hidden in neighboring $PM_{2.5}$ monitoring sites. Then, the extracted features are connected with a random forest model for $PM_{2.5}$ estimation. In addition, we

verify the effect of AOD data quality on the final estimation accuracy of the model and analysis of the feature importance ranking results to form some explanatory notes on the estimation model. Finally, PM$_{2.5}$ estimations were formed by fusing satellite and model data, and we make a comparative analysis of spatial–temporal information at different scales to further demonstrates the rationality of the estimated results.

## 2. Materials and Methods

### 2.1. Datasets and Preprocessing

The experimental data in this paper include the hourly PM$_{2.5}$ observations from monitoring stations, 5 km resolution AOD products of Himawari-8, and other auxiliary data. Table 1 lists all the information about the data used in this study. The study area in this paper is the Beijing–Tianjin–Hebei urban agglomeration, with the geographical location of 113.45° E 119.83° E and 36.08° N 42.67° N. Figure 1 shows the geographical location of the region and the distribution of the state-controlled environmental monitoring sites located within the region. The elevation map in the figure shows that the topographic distribution of the region has its own distinctive characteristics. The Beijing–Tianjin–Hebei urban agglomeration has a high population concentration and is one of the core economic regions in China. In addition, with the air pollution control in recent years, the spatial pattern of some major steel, coal, power, and light industries in the Beijing-ring region has changed dramatically. As one of the heavy industrial regions in China, most of the industrial production has shifted to the southern part of Hebei province, which further leads to the clustering of heavy industry in this areas. In terms of topography and climatic environment, air pollution tends to accumulate in these regions. In terms of pollution sources, the periphery of the region is a heavy industrial agglomeration, which has a certain impact on the air quality of Beijing in the central region of the urban agglomeration.

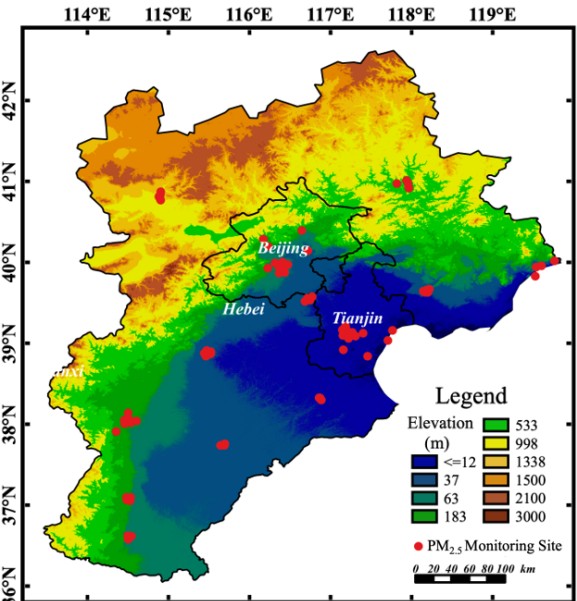

**Figure 1.** Geographical location of the study area and distribution map of PM$_{2.5}$ monitoring sites.

2.1.1. Ground-Level PM$_{2.5}$ Measurements

Hourly PM$_{2.5}$ observations from monitoring stations in BTH of 2017 are collected for model fitting and validation. They are publicly available on the website of the China Environmental Monitoring Centre (CEMC, http://www.cnemc.cn/, accessed on 7 February 2024). As shown in Figure 1, there are about 80 monitoring sites in total, and most of them are distributed in the city areas of BTH.

**Table 1.** Summary of the dataset.

| Type | Abbreviation | Content | Spatial Resolution | Temporal Resolution | Source |
|---|---|---|---|---|---|
| PM$_{2.5}$ | PM$_{2.5}$ | PM$_{2.5}$ | site | hourly | CNEMC |
| AOD | H-AOD | Himawari AOD | 5 km | hourly | JAXA |
| | M-AOD | Himawari model AOD | 5 km | hourly | JAXA |
| Meteorological | TEM | 2 m air temperature | 0.1° × 0.1° | hourly | ECMWF |
| | UW | 10 m u-component of wind | 0.1° × 0.1° | hourly | ECMWF |
| | VW | 10 m v-component of wind | 0.1° × 0.1° | hourly | ECMWF |
| | PRE | Total precipitation | 0.1° × 0.1° | hourly | ECMWF |
| | SP | Surface pressure | 0.1° × 0.1° | hourly | ECMWF |
| | BLH | Boundary layer height | 0.25° × 0.25° | hourly | ECMWF |
| | RH | Relative humidity | 0.25° × 0.25° | hourly | ECMWF |
| Land-related | NDVI | NDVI | 1 km | 16-day | MYD13A2 |
| | DEM | DEM | 90 m | - | SRTM |
| | LULC | LULC | 500 m | annually | MCD12Q1 |

### 2.1.2. AOD Data

Himawari-8 is a Japanese geostationary satellite that launched on 7 October 2014. It carries an Advanced Himawari Imager (AHI) and produces hourly AOD at a 5 km resolution during the daytime. In recent years, Himawari-8 AOD has been evaluated and used increasingly in the study of China air pollution [21,22,44]. Here, we obtained official Himawari-8 AOD (Level 3 Version 3.0) at 500 nm in 2017 and used AOD-merged data, which are with broader spatial and temporal coverage and slightly better quality [45]. Meanwhile, model-assimilation AOD products of Himawari-8 were selected for achieving full-coverage PM$_{2.5}$ estimation. This product is the forecast (every hour) of aerosol properties by the MRI/JMA global aerosol model called the Model of Aerosol Species in the Global Atmosphere (MASINGAR). This product is assimilated by Himawari L3 aerosol optical depth at 00, 03, 06, and 09 UTC. There are quality differences between these AOD data, which could help to verify the effect of AOD data quality on the final PM$_{2.5}$ estimation accuracy of the model.

### 2.1.3. Auxiliary Data

Auxiliary data consist of meteorological and land-related data. They are helpful for capturing the complex AOD-PM$_{2.5}$ relationships associated with spatiotemporal variations. The meteorological data are extracted from the European Centre for Medium-Range Weather Forecasts (ECMWF, https://www.ecmwf.int, accessed on 7 February 2024) and ERA5-Land reanalysis products, including temperature (TEM; unit: K), surface pressure (SP; unit: hPa), relative humidity (RH; unit: %), precipitation (PRE; unit: mm), and 10m u-component and v-component of wind (UW/VW; unit:m/s). The boundary layer height (BLH; unit: m) is obtained from the ERA5 hourly data on single levels. In addition, this study adopted three land-related datasets, including the normalized difference vegetation index (NDVI) products derived from the MODIS MYD13A2, the digital elevation model (DEM) products inferred from the shuttle radar topography mission, and the land use/cover (LULC) products derived from the MODIS MCD12Q1.

### 2.1.4. Processing of Data

Although the spatial and temporal resolution of current satellites has been greatly improved, it is difficult for a single data source to meet the high-precision requirement of all applications. Moreover, due to the influence of cloud coverage, complex surface conditions, and other factors, satellite AOD data often also have a large amount of spatial information missing. In order to obtain full-coverage of PM$_{2.5}$ spatial and temporal distribution data, the research on the reconstruction of missing information in satellite remote sensing esti-

mated $PM_{2.5}$ can be summarized into two main aspects [8,17,31]: one is to prioritize the reconstruction of AOD missing data, so as to reduce or eliminate the spatial missingness of estimated $PM_{2.5}$ data; the others is to directly use single-source satellite AOD data for $PM_{2.5}$ estimation and then seamlessly reconstruct it by fusing multiple sources of $PM_{2.5}$.

In order to achieve $PM_{2.5}$ estimation at 1 km, we improved the official Himawari-8 hourly AOD at a 5 km resolution to 1 km by using the AeroCGAN model [46]. Meanwhile, estimation experiments were conducted separately using three types of AOD data: official Himawari-8 AOD (H-AOD), official Himawari-8 model-AOD (M-AOD), and mixed-AOD (Mix-AOD) of the above. Then, according to different AODs with ground-based monitoring $PM_{2.5}$, we make a temporal and spatial matching processing for AODs and relevant auxiliary data, and then form a control variable experiment dataset. In this way, we will verify the effect of AOD data quality on the final estimation accuracy of the model and achieve full-coverage $PM_{2.5}$ estimation.

In addition, since the auxiliary data are diverse in terms of spatial resolutions, they are resampled to a 1 km resolution with bilinear interpolation to ensure data consistency. Bilinear interpolation can produce smooth interpolation results with the distance-weighted average of the four nearest pixels.

After data processing, based on the data in the study area in 2017, there are 115,029 valid samples matched for satellite observation AOD data, and 222,547 valid samples data matched for both model-assimilation AOD data and mixed AOD data. These data will be used for model training and validation.

### 2.2. Methods

Supported by abundant observational data and machine learning methods, the main current $PM_{2.5}$ estimation methods are essentially optimization processes that are searching for the relationship between $PM_{2.5}$ concentrations and observational data. More formally, the commonly used model can be expressed as $PM_{2.5} = f$ (AOD, meteorological, land-related, geographic factors).

Typically, the AOD is essential, basic data. Meteorological factors and land-related data are listed in Table 1; they are helpful for capturing the complex AOD-$PM_{2.5}$ relationships associated with spatiotemporal variations. In addition, geographic factors, such as the neighboring ground-based $PM_{2.5}$ measurements, are valuable prior information for spatial correlations. Previous studies measured this correlation by weighting spatio-temporal distance [19,37]. However, as shown in Figure 2, ground-based $PM_{2.5}$ monitoring sites are usually sparsely distributed in urban areas. Moreover, such correlations are usually dynamic and highly susceptible to changes in time and local environmental conditions. When the correlations are constructed on the basis of spatial distance [19,37], in most studies, the weight of such correlations is not consistent with the spatial and temporal heterogeneity of $PM_{2.5}$. For example, when several neighboring stations with similar spatial distances are in the upstream and downstream channels of pollution dispersion, it is obvious that the upwind stations are less affected, but the downwind stations are relatively more affected. It is difficult to distinguish their differences with the correlation only based on distance.

According to the above analysis, common weight methods with spatio-temporal distance show strong effectiveness in spatio-temporal correlation information when training the model with training data (like $P_1$), but they provide weak effectiveness when actually applied to large-area data (like $P_2$), as in Figure 2.

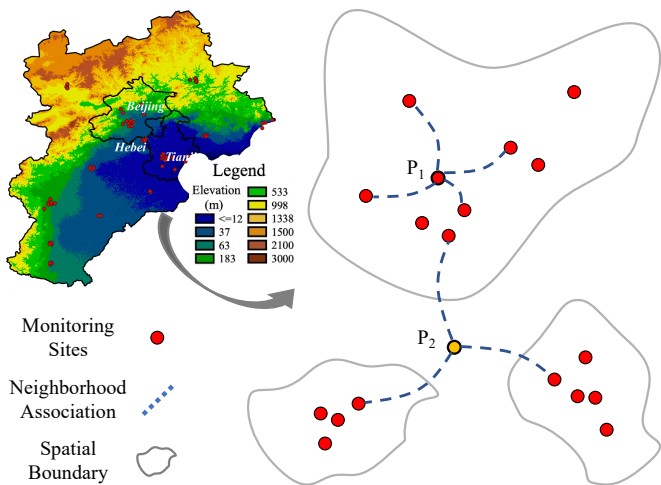

**Figure 2.** Schematic diagram of the spatial–temporal correlation of PM$_{2.5}$ ground monitoring stations.

### 2.2.1. Spatial Features

In order to more efficiently utilize the spatially correlated prior information provided by the observations of neighboring sites, we construct a co-weighting similarity of spatial distance and environment factors using the attention mechanism. Attention mechanism-based neural network models have been widely used in computer vision, natural language processing, and spatio-temporal prediction tasks [47,48]. Essentially, the attention mechanism is used to calculate the similarity relationship of the vectors and learn the features of the data accordingly. Inspired by this mechanism, we designed the new spatial feature extraction method.

For the PM$_{2.5}$ concentration of a target pixel, its neighboring PM$_{2.5}$ sites usually show a certain correlation to it. Just like in the first law of geography, all things are related, but nearby things are more related than distant things [49]. This law can interpret previous studies that constructed this correlation by weighted spatio-temporal distances [19,37]. However, the impact of weights are highly dynamic, when changing over time and with environmental factors. Therefore, this study further considers the third law of geography: the more similar the geographic configurations of two points (areas), the more similar the values (processes) of the target variable are at these two points (areas) [50]. With this law, we construct co-weights of spatial distance (the first law) and environmental similarity (the third law) by using an attention mechanism. Moreover, since there might be many irrelevant series, it results in very high computational cost and degrades the performance if directly using all kinds of time series as the encoder inputs to capture the correlations between different observation data. Therefore, we further synergize the environmental similarity and spatial distance to capture the dynamic correlations by using a designed attention mechanism.

Let $s_{target}$ be the site of our estimated target, and its neighboring sites are $S = [s_1, \cdots, s_i, \cdots, s_N]$, every site $s_i$ has an environment vector $\mathbf{e}_i$ and a PM$_{2.5}$ value $p_i$.

$$\mathbf{e}_i = [AOD, TEM, RH, BLH, WS, WD, NDVI] \tag{1}$$

where AOD, TEM, RH, BLH, WS, WD, and NDVI are defined in Table 1. Furthermore, $s_{target}$ has an environment vector $\mathbf{e}_{target}$ like $\mathbf{e}_i$. To reduce the computational complexity of the attention mechanism in extracting spatially correlated features, only these seven variables were selected for the environmental similarity extraction. Then, we calculate the attention weights Sim$_i$ (i.e., impacting weight) between neighboring sites as follows:

$$\text{Sim}_i = \boldsymbol{\eta} \tanh\big(\mathbf{W}_i\big[\mathbf{e}_i; \mathbf{e}_{target}\big] + \mathbf{K}_i\mathbf{e}_{target} + \mathbf{b}_i\big) \tag{2}$$

where $\left[\mathbf{e}_i; \mathbf{e}_{target}\right]$ is a concentration operation between $\mathbf{e}_i$ and $\mathbf{e}_{target}$. $\boldsymbol{\eta}$, $\mathbf{W}_i$, $\mathbf{K}_i$, and $\mathbf{b}_i$ are the learnable parameters.

Then, a SoftMax function operation is performed to normalize all attention weights. In this process, we construct the neighboring site attention weights considering both spatial distance and environmental similarity.

$$a_i = \text{Softmax}(\text{Sim}_i) = \frac{\exp\left((1-l_i)\,\text{Sim}_i + l_i\right)}{\sum_{j=1}^{N}\exp\left((1-l_j)\,\text{Sim}_j + l_j\right)} \tag{3}$$

where $l_i$ is the geospatial distance weight, and the specific value is the reciprocal of the spatial distance $\left(\frac{1}{d_i}\right)$ between the site target site $s_{target}$ and neighboring site $s_i$. Similarity $a_i$ indicates the attention value between the *i*-th neighboring site and the site to be estimated.

The attention weights of spatial features are jointly determined by the input features and the spatial distance in the encoder. The score calculated by Equation (3) semantically represents the site correlation under the effect of distance and environment. When the spatial distance between two sites is closer, the effect of spatial distance weight is higher than the environmental similarity weight, thus the correlation between them is higher; on the contrary, when the spatial distance between two sites is farther, the effect of the environmental similarity weight is higher than that of the spatial distance weight, thus the environmental similarity effect is higher.

Finally, we weighted the spatial correlation of $PM_{2.5}$ for all spatially neighboring sites and obtained the neighboring spatial features $\text{SAtt}_{s_{target}}$ of the site to be estimated as follows:

$$\text{SAtt}_{s_{target}} = \sum_{i=1}^{n} a_i \cdot p_i \tag{4}$$

where $a_i$ is the spatial distance and environmental similarity co-weighting by Equation (3), and $p_i$ is the $PM_{2.5}$ concentration at the *i*-th site.

In addition, considering the numerical differences among the variables, each variable was normalized separately before feature extraction.

### 2.2.2. Temporal Features

The time stamp could be a temporal feature in some machine learning methods; especially, the minimum data unit of this study is a vector of features. Moreover, numerous studies have verified that the values of $PM_{2.5}$ concentrations show strong variation and correlation patterns across the geographical space and the temporal dimension [22,51,52]. The sites in the BTH often observe much higher $PM_{2.5}$ concentrations in winter than summer. The $PM_{2.5}$-AOD correlation became noticeably higher from 9:00 to 17:00 local time, whereas the $PM_{2.5}$/AOD ratio notably decreased in the Beijing–Tianjin–Hebei, Yangtze River Delta, and Chengyu regions. Specifically, there is a high correlation between 12:00 and 14:00 LT (Local Time) and 13:00 and 17:00 LT. The ratio in a day has a clear unimodal pattern, with the peak occurring, particularly in the fall and winter, at about 10:00 or 11:00 LT. The $PM_{2.5}$-AOD association significantly varies over the course of a week in the winter. Additionally, the winter is the period of time when most metropolitan agglomerations have their best association and highest ratio [53]. Therefore, considering that our approach aims to estimate hourly $PM_{2.5}$ concentrations, we selected month and hour as temporal features.

### 2.2.3. Random Forest

Random forest is based on the idea of bootstrap aggregating and the random subspace method [39]. It aggregates a set of weak learners to form a strong one; these weak learners, i.e., basic decision trees, are trained by randomly bootstrapped samples from the training set. Moreover, the node splitting features are also randomly selected during the training process of the decision tree. Thus, it can nonparametrically model the indefinable and complex non-linear relationship among input features [11,19].

Decision tree-based models, such as RF, XGBoost, and LightGBM, are more advantageous in fitting data with tabular features as well as PM$_{2.5}$ estimating. Previous studies have demonstrated that there is little difference in performance between these decision tree-based models in PM$_{2.5}$ estimation [38]. Compared to image or natural language data, tabular data are heterogeneous, resulting in dense numbers and sparse categorical features. In addition, the correlation between features is weaker than the spatial or semantic relationships in image or natural language data. The variables of tabular data can be correlated or independent, and features usually have no positional information. Therefore, it is necessary to discover and exploit correlations relying on spatial information. The spatio-temporal correlation feature extraction is the primary optimization direction of these PM$_{2.5}$ estimation models.

Especially, PM$_{2.5}$ concentrations show strong variation and correlation patterns across the geographical space and the temporal dimension. It is difficult to capture such features by relying on random forest alone to mine the value of neighboring site data. Therefore, this study selected the classic random forest model and aggregating attention mechanism to achieve PM$_{2.5}$ estimation. With the attention mechanism described above, we obtain new spatio-temporal features that are fed into the random forest together with the auxiliary data.

There are three key parameters that should be tuned in a random forest model, which are the number of trees in the forest (N), the number of input features to consider when splitting data at a decision node (m), and the minimum number of samples required to be at a leaf node (n). Through controlled variable experiments, we finally set each of these three parameters as 500, 13, and 2, respectively. Actually, this is also related to the amount of our data and the number of features.

### 2.2.4. Model and Evaluation

According to the above analysis, the flowchart of the proposed attention mechanism-based random forest model is shown in Figure 3, including input data, feature processing, RF regression, and output data. Firstly, we made a temporal and spatial matching processing for AODs and relevant auxiliary data to form a training dataset. Then, we used the attention mechanism-based spatial correlation extraction method (Equations (1)–(4)) to obtain the spatial feature. Furthermore, the time stamp is chosen as the temporal feature. These above data are used as input features of the random forest model to finally estimate PM$_{2.5}$ concentrations. Here, the steps of the attention mechanism-based random forest are summarized in Algorithm 1.

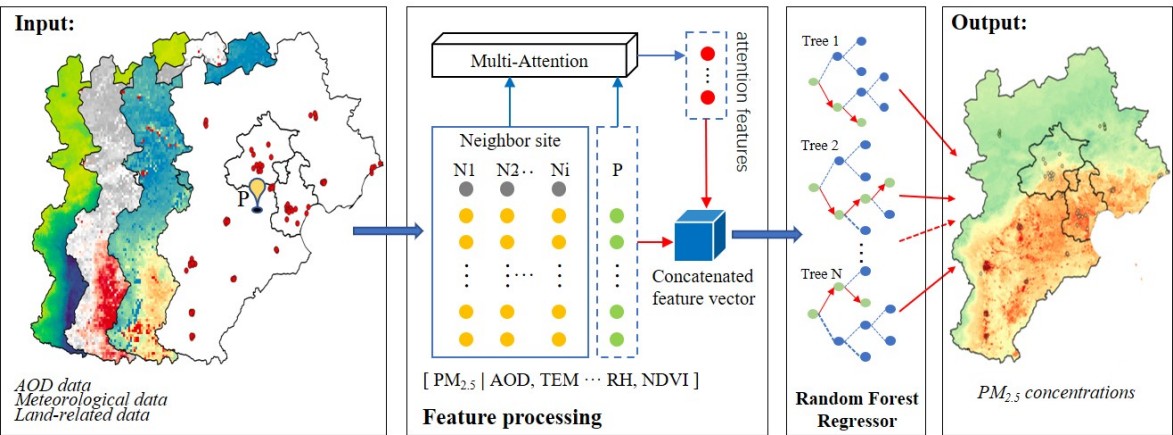

**Figure 3.** Attention mechanism-based random forest model for PM$_{2.5}$ concentration estimation.

---

**Algorithm 1** Attention Mechanism-Based Random Forest Algorithm for PM$_{2.5}$ Estimation

---

**Input:** Dataset $D = \{(x_1, y_1), \cdots, (x_i, y_i), \cdots, (x_N, y_N)\}$, each $x_i$ contains $F$ features (defined in Table 1). Training times is $T$. The random forest RF has $M$ decision trees (DTree).

1: SAtt ← Eqs. (1) to (4); attention mechanism-based spatial correlation
2: TF; time stamp as temporal features
3: $x_i' \leftarrow [x_i, \text{SAtt}_i, \text{TF}_i]$; features concat
4: $D' = \{(x_1', y_1), \cdots, (x_i', y_i), \cdots, (x_N', y_N)\}$; each $x_i'$ contains $F'$ features (i.e., dimension);
5: **for** $t = 1, 2, \ldots, T$ **do**
6:     randomly bootstrapped K samples:
7:     $D_1, \cdots, D_k, \cdots, D_K \in D'$
8:     **for** $D_k \in D'$ **do**
9:         Randomly select $f$ feature subsets from the $F'$-dimension features;
10:        training each decision tree:
11:        **for** DTree$_i$ in RF **do**
12:            select the optimal one from the $f$ features for tree node splitting
13:            DTree$_i \leftarrow \{D_k, f\}$
14:        **end for**
15:    **end for**
16:    RF$_t \leftarrow \{\text{DTree}_1, \cdots, \text{DTree}_i, \cdots, \text{DTree}_M\}$
17: **end for**
18: **return** RF

---

In addition, based on previous studies [8,19,28], 10-fold cross-validation [54] is adopted to investigate the effectiveness of the proposed method. The general 10-fold cross-validation randomly partitions all the samples into 10 folds with an equal number of subsamples, which is also referred to as sample-based cross-validation. One subsample is retained for validation, and the remaining 9 subsamples are used for training. This process of subsampling is repeated 10 times with each fold for validation in return. The overall performance of the model is estimated by averaging the 10 evaluation results to reduce the random effect.

During the evaluation stage, three quality metrics are used to measure the performance: determination coefficient ($R^2$), root-mean-square error (RMSE), and mean absolute error (MAE). They are denoted as follows:

$$R^2 = \frac{\sum_{i=1}^{n}(P_i - \bar{Q}_i)^2}{\sum_{i=1}^{n}(O_i - \bar{Q}_i)^2} \tag{5}$$

$$RMSE = \sqrt{\frac{1}{n}\sum_{i=1}^{n}(P_i - Q_i)^2} \tag{6}$$

$$MAE = \frac{1}{n}\sum_{i=1}^{n}|P_i - Q_i| \tag{7}$$

where $O_i$ is the obvious value, $\bar{Q}_i$ is the average of obvious values, $P_i$ is the estimate value, and $n$ is number of data points.

## 3. Results and Discussion

### 3.1. Model Performance

Table 2 shows the estimation performances of the proposed and some existing methods. With the development of analysis methods, the early geographically weighted regression model (GWR) is gradually replaced by some new methods that are based on machine learning (XGBoost, Two-stage, STRF). Furthermore, the stream trend of the current research is to further improve the performance of the machine learning method by adding spatio-temporal features. In addition, it can be seen that the accuracy for most of the

model estimation is within a certain range ($R^2$: 0.65 to 0.90; RMSE: 10.0 to 30.0 μg/m$^3$, approximately). This may be due to the difference in the spatial and temporal resolution of the data or the spatial and temporal range of the research. For example, some early studies for PM$_{2.5}$ estimation [14,37] are mostly based on 10 km resolution AOD data from MODIS satellite observations. Then, to benefit from the improvement in AOD data quality (such as MODIS MAIAC AOD with 1 km spatial resolution, Himawari AOD with 5 km hourly resolution, etc.) and the development of machine learning methods, some researchers have gradually developed more efficient PM$_{2.5}$ estimation methods [19,22,31], and the study area has changed from highly polluted areas (such as Beijing–Tianjin–Hebei, Yangtze River Delta, and Pearl River Delta) to large regional nationwide areas and long time series.

**Table 2.** Comparison of performance with other methods.

| Methods | Resolution | $R^2$ | RMSE | MAE | Source AOD | Period | Region | Reference |
|---|---|---|---|---|---|---|---|---|
| LME | 10 km, daily | 0.79 | 26.74 | - | MODIS | 2015 | China | Ma et al. (2016) [27] |
| GWR | 10 km, daily | 0.64 | 32.98 | 21.25 | MODIS | 2013 | China | Ma et al. (2014) [14] |
| GTWR | 3 km, daily | 0.80 | 18.00 | 12.03 | MODIS | 2015 | China | He et al. (2018) [28] |
| Geo-DBN | 10 km, daily | 0.88 | 13.03 | 8.54 | MODIS | 2015 | China | Li et al. (2017) [37] |
| DNN | 1 km, hourly | 0.84 | 19.90 | 11.89 | Himawari | 2017 | BTH | Sun et al. (2019) |
| Two-stage | 1 km, daily | 0.85 | 11.02 | - | MODIS, Himawari | 201804–201902 | China | Jiang et al. (2021) [26] |
| STRF | 1 km, daily | 0.85 | 15.57 | 9.77 | MODIS MAIAC | 2015–2016 | China | Wei et al. (2019) [19] |
| STET | 1 km, daily | 0.89 | 10.35 | 6.71 | MODIS MAIAC | 2017–2018 | China | Wei et al. (2020) |
| STLG | 5 km, hourly | 0.85 | 13.09 | 8.11 | Himawari | 2018 | China | Wei et al. (2021)[22] |
| XGBoost | 5 km, hourly | 0.84 | 18.10 | 11.40 | Himawari | 2016 | Central and Eastern China | Chen et al. (2019) [38] |
| RF | 1 km, hourly | 0.81 | 25.51 | 15.91 | Himawari | 2017 | BTH | This study |
| STAttenRF | 1 km, hourly | 0.89 | 18.31 | 11.17 | Himawari | 2017 | BTH | This study |

In this study, we use the attention mechanism to construct spatial proximity feature extraction and then introduce it into the random forest model for PM$_{2.5}$ estimation. We also compared it with the performance of the basic random forest model without spatio-temporal features ($R^2$ = 0.81, RMSE = 25.51 μg/m$^3$, MAE = 15.91 μg/m$^3$). The proposed model which captures the spatio-temporal features, achieves a higher efficiency ($R^2$ = 0.89, RMSE = 18.31 μg/m$^3$, MAE = 11.17 μg/m$^3$). Although the accuracy index of the proposed method is not the highest among the existing methods, it has definite improvements in spatial and temporal resolution (1 km, hourly). Meanwhile, it is also demonstrated that the spatial proximity features extracted by using the attention mechanism can provide effective supports for the final estimation.

### 3.2. Feature Correlation and Importance Analysis

For the task of estimating PM$_{2.5}$ concentrations, most machine learning methods are black-box models which do not provide us with much knowledge about the massive data. In other words, they do not tell us how these input variables are connected to the output of the predictions and which factor gains the most attention in the model. In previous studies, they usually used the Pearson correlation coefficient and histograms as descriptive statistics for the correlation analysis [20,53]. Therefore, this paper also conducts a preliminary feature correlation analysis for the problem accordingly.

The frequency distribution histograms of the data with descriptive statistics (minimum, maximum, mean, and standard deviation) are shown in Figure 4. It can be seen that the frequency distribution of PM$_{2.5}$ concentration is more similar to the AOD data, which is in line with the consensus of much research that the AOD data can directly reflect the corresponding spatial and temporal distribution characteristics of PM$_{2.5}$ concentration, and they have a certain degree of similarity in spatial and temporal distribution. In addition, there is a large difference between the Himawari satellite retrieval AOD and the model-assimilation AOD. The value of model-assimilation AOD data is higher than that of the satellite inversion overall, which indicates that the model-assimilation AOD value

may be overestimated, but the overall frequency distribution structure has some similarity between these data. Although some previous studies have shown that the relative humidity, air temperature, and wind speed have a strong influence on the spatial and temporal distribution of PM$_{2.5}$ concentrations [8], it is difficult to show the intuitive similarity only in their frequency distributions. These results also reflect that the relationship among the spatio-temporal distribution of PM$_{2.5}$ concentrations, meteorological factors, and surface environment data is complex and nonlinear. The DEM data are sparsely distributed because the data are obtained by matching the air quality monitoring stations and the DEM values reflecting the geospatial location of the stations to a certain extent.

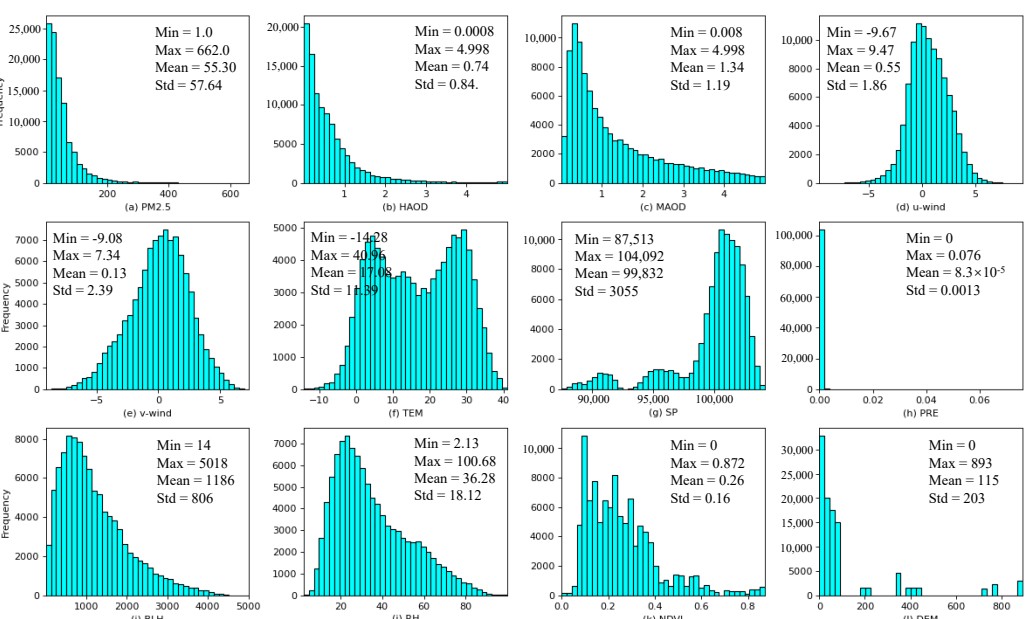

**Figure 4.** Histograms and descriptive statistics (minimum, maximum, mean, and standard deviation) of PM$_{2.5}$ concentrations and the associated variables.

The frequency distribution of PRE data is very concentrated and near the Y axis; it may have a small effect on the accuracy of PM$_{2.5}$ estimation in the model and needs to be further confirmed for it to be retained in the subsequent experimental analysis. According to the above analysis, to ensure the efficiency of the PM$_{2.5}$ estimation model, suitable characteristic variables need to be selected, and the nonlinear relationships between multiple variables need to be adequately fitted and to be experimentally compared to form a definitive model data selection strategy.

The Pearson correlation coefficients between PM$_{2.5}$ concentration data and other variables are shown in Figure 5. It can be seen that the Pearson coefficient of PM$_{2.5}$ concentration and Himawar satellite AOD (H-AOD) data reaches 0.43, which is the highest among all variables, and this further indicates that the AOD is always the core data for PM$_{2.5}$ estimation. In addition, among the variables related to the atmospheric physical properties of PM$_{2.5}$, the correlation between PM$_{2.5}$ concentration and boundary layer height and relative humidity is higher (both reach 0.31), while the correlation with wind speed ($-0.16$, $0.14$) and temperature ($-0.061$) have lower correlations. Usually, PM$_{2.5}$ pollution is more influenced by wind speed and temperature, etc., during dissipation, and this single intuitive coefficient makes it difficult to reflect the characteristic correlation of PM$_{2.5}$ concentration. Therefore, although the histogram structure of frequency distribution of AOD and PM$_{2.5}$ data is similar and the Pearson correlation coefficient is high, the accuracy of directly constructing the fitted relationship between AOD and PM$_{2.5}$ for PM$_{2.5}$ estimation is limited, and it is necessary to use multivariate synergy to fit the nonlinear relationship.

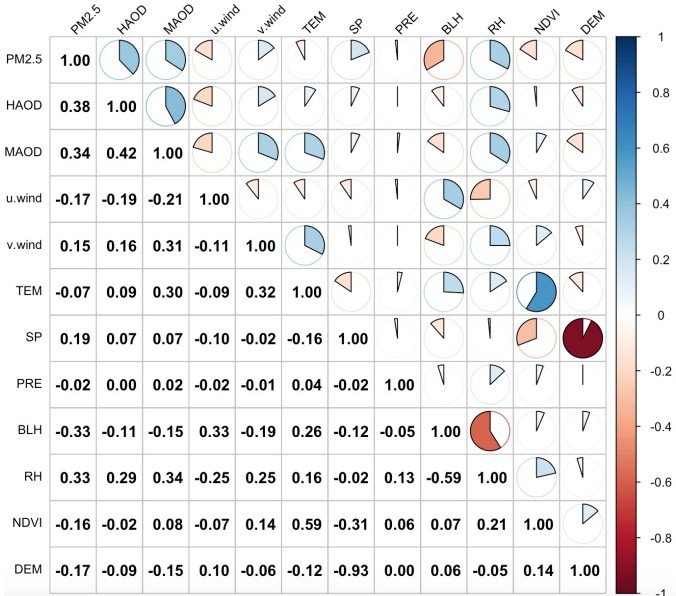

**Figure 5.** The results of bivariate correlation analysis between PM$_{2.5}$ concentrations and other associated variables.

The random forest model requires a tree node splitting calculation during the construction of the decision tree, such as a splitting function based on the gini coefficient. This approach also reflects the importance of the feature variables on the estimation of the random model. Therefore, this study provides a detailed comparative analysis of the importance of information related to these variables in the random forest, as shown in Figure 6. Overall, among the feature variables, the model considers AOD, BLH, and RH to be the most important for the estimation results, which is also consistent with most existing studies [19,55]. Moreover, this information importance also corresponds in some aspects to Figures 4 and 5. When the model added temporal features, the time stamp (Month, Day, Hour) shows a certain of importance for the estimation results. There is little change in the feature importance ranking, but only the NDVI importance rate is reduced greatly.

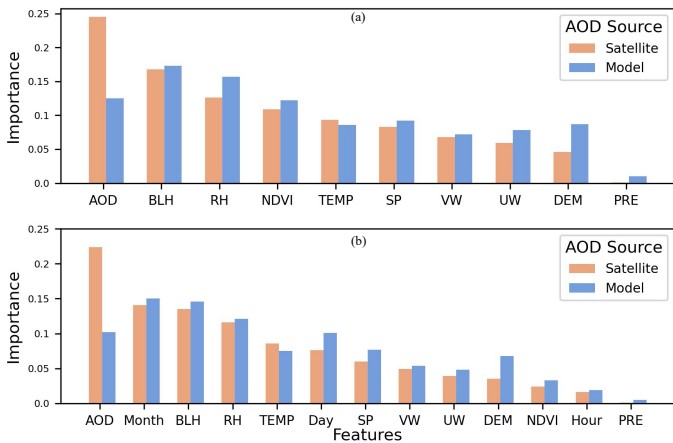

**Figure 6.** Feature importance comparison of input variables in PM$_{2.5}$ concentrations estimation model. (**a**) RF model. (**b**) STAttenRF model

### 3.3. The Impact of AOD data Quality on Model Accuracy

To further compare and analyze the effect of AOD data quality on the estimation of atmospheric PM$_{2.5}$ by the random forest model, we conducted separate comparison experiments according to the different AOD data sources, as shown in Table 3. Obviously, the coverage of model assimilation AOD data is higher. In addition, we verified the correlation

accuracy (satellite retrieval and model assimilation AOD data vs. the AOD data from ground monitoring sites), as shown in Figure 7. There are 6169 valid data matched between the satellite observed AOD data and AERONET in the study area in 2017, and the correlation between them reached 0.836. There are 6946 valid data matched between the model assimilation AOD data and AERONET, and the correlation between them was only 0.585. It is shown that there is a huge difference in data quality between the two cases. Furthermore, the visual comparison of the scatter plotted in Figure 7 also shows that there is a large overestimation of AOD in the model assimilation AOD. However, when these AOD data are input into the random forest model to estimate the spatial and temporal distributions of $PM_{2.5}$, Table 3 shows that the difference in estimation results is small, as well as the $PM_{2.5}$ estimation based on the Himawari satellite retrieval AOD being $R^2 = 0.812$, RMSE = 25.51 $\mu g/m^3$, and MAE = 15.91 $\mu g/m^3$, and the Himawari model assimilation AOD being $R^2 = 0.805$, RMSE = 27.86 $\mu g/m^3$, and MAE = 15.35 $\mu g/m^3$. These results are somewhat different from previous studies that pursued the quality of AOD data and used it to obtain high quality $PM_{2.5}$ estimation. However, by mixing the two types of AOD data and replacing them with model assimilation AOD data when the AOD is missing from the satellite retrievals, the accuracy of the model is reduced to a certain extent. Based on this comparison, this study concluded that there may be errors in the full-coverage $PM_{2.5}$ estimation by directly using the mixed AOD, so the full-coverage results should be obtained by estimating $PM_{2.5}$ separately and then performing data fusion. The full-coverage results were obtained by fusing the data after estimating $PM_{2.5}$ separately. In addition, after using the attention mechanism to extract spatio-temporal features, the model accuracy was significantly improved. As shown in Table 3, the evaluation indicators of $PM_{2.5}$ estimation by Himawari satellite retrivals AOD were $R^2 = 0.887$, RMSE = 18.31 $\mu g/m^3$, and MAE = 11.17 $\mu g/m^3$, and the index of $PM_{2.5}$ estimation by Himawari model assimilation AOD was $R^2 = 0.874$, RMSE = 20.68 $\mu g/m^3$, and MAE = 12.87 $\mu g/m^3$. These situations further illustrate the importance of spatio-temporal characteristics for the random forest model.

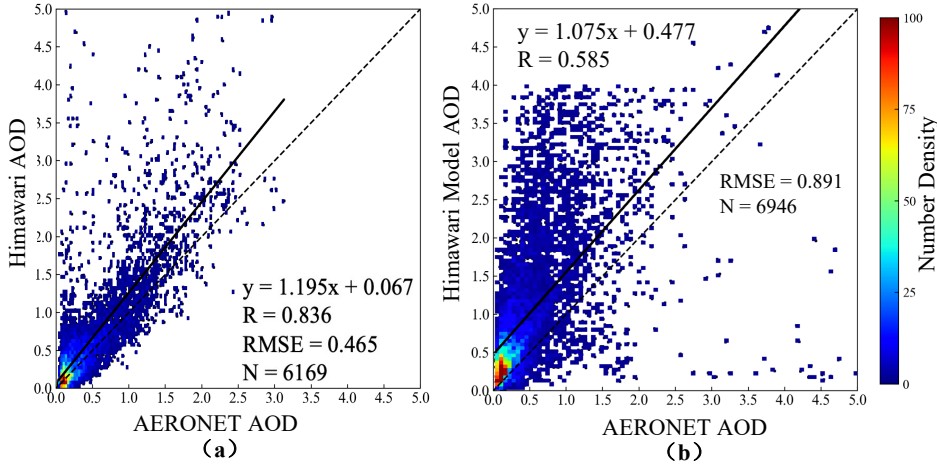

**Figure 7.** Comparison results of ground-based validation of Himawari (**a**) satellite retrievals and (**b**) model assimilation of AOD data.

With the comparison of the importance of input variables in the random forest model in Figure 6, we provide a preliminary explanation for this phenomenon. Although the correlation between AOD and $PM_{2.5}$ is strong, the random forest responds differently to AOD data of different quality when multiple variables are input into it. For example, for the Himawari AOD data in Figure 6, the random forest considered the AOD data to be the most important (more than 20%). While for the model-assimilation AOD data, for its lower data precision (Figure 7), the random forest model adaptively reduced its importance of layer height BLH and relative humidity RH. However, the model is still able to generate better $PM_{2.5}$ spatial and temporal distribution estimation results in cooperation

with several variables in the end. In the estimation results by the mixed AOD data, all three evaluation indicators show some degree of degradation compared with the separate estimation. This may be due to the misleading effect of data quality differences on the same characteristic variables in the random forest model, which is difficult to distinguish in the random forest model.

Based on above analysis, we believe that although the quality of AOD data varies greatly, when multiple variables are input into the random forest model, the random forest will adjust the importance of different-quality AOD data to achieve a certain of robustness (the $PM_{2.5}$ distribution estimated based on satellite retrievals AOD data or model-assimilation AOD data). The non-linear relationship between the variables can be fitted well, and thus, the spatial and temporal distribution of $PM_{2.5}$ can be estimated better. In addition, comparing the final results, we can find that the model accuracy is significantly improved after using the attention mechanism to extract spatio-temporal features.

**Table 3.** Comparison of experimental results of $PM_{2.5}$ estimation with different AOD data

| Data | Method | Model Fitting | | | Model Validation | | |
|---|---|---|---|---|---|---|---|
| | | $R^2$ | RMSE | MAE | $R^2$ | RMSE | MAE |
| H-AOD | RF | 0.972 | 9.45 | 5.46 | 0.812 | 25.51 | 15.91 |
| | STAttenRF | 0.983 | 7.56 | 4.51 | 0.887 | 18.31 | 11.17 |
| M-AOD | RF | 0.973 | 9.94 | 5.66 | 0.805 | 27.86 | 15.35 |
| | STAttenRF | 0.985 | 7.47 | 4.34 | 0.874 | 20.68 | 12.87 |
| Mix-AOD | RF | 0.968 | 10.18 | 5.96 | 0.793 | 28.20 | 17.02 |
| | STAttenRF | 0.981 | 7.63 | 4.54 | 0.861 | 22.43 | 13.66 |

*3.4. Spatio-temporal Validation and Analysis*

The spatial and temporal distribution of seasonal $PM_{2.5}$ concentration is calculated from the average of hourly $PM_{2.5}$ data in the corresponding time interval, as shown in Figure 8. Among them, the point source distribution markers in the figure are the seasonal averages of the ground monitoring stations, which range from March to May each year in spring, June to August each year in summer, September to November each year in autumn, and December to February each year in winter. It can be seen that the seasonal distribution of $PM_{2.5}$ concentrations in the Beijing–Tianjin–Hebei region varies significantly. Based on the statistical data of $PM_{2.5}$ pollution concentrations at ground stations, it can be concluded that the pollution level is higher in winter ($\sim$70.12 $\pm$ 85.26 µg/m$^3$), the average values in spring ($\sim$47.61 $\pm$ 46.09 µg/m$^3$) and autumn ($\sim$46.94 $\pm$ 42.92 µg/m$^3$) are close to each other, and the lowest pollution level is in summer ($\sim$42.15 $\pm$ 25.23 µg/m$^3$). The seasonal $PM_{2.5}$ spatial distribution data were obtained statistically after model estimation in this section, and the $PM_{2.5}$ pollution level in the northwest region of Beijing, Tianjin, and Hebei is lower. The $PM_{2.5}$ pollution level in the southeast region is higher, which shows a trend of gradually severe pollution from the northwest to southeast. This spatial pattern distribution is extremely similar to the digital elevation map of the Beijing–Tianjin–Hebei region shown in Figure 1. With higher terrain in the northwest mountainous region and lower terrain in the southeast open plain, it shows a sloping trend from the northwest to southeast. It can be seen that $PM_{2.5}$ pollution is mainly distributed in the urban built-up areas, while decreasing spatially toward the suburbs. The main reasons are: the northwest region is located in the mountainous area, the population and urban built-up areas are sparsely distributed, and there are few sources of pollution emissions caused by industrial agriculture and human activities. In contrast, Beijing–Tianjin–Hebei is one of the core economic regions in China and also one of the heavy industrial regions in China. With the air pollution control in recent years, the spatial pattern of some major iron and steel, coal, electricity, and light industries in the Beijing-ring region has changed dramatically. Most of the industrial sites have shifted to the southern part of Hebei province, thus leading to poorer air quality in

the southeast, which is the area of heavy industries. Although the pollution sources around Beijing are reduced, topographically, the pollution from the southeast heavy industry region still spreads to the Beijing area and gathers and dissipates in the region. Beijing is still a population gathering area, and it is difficult to reduce the traffic as well as living pollutant emissions, especially the large amount of pollution emissions caused by coal burning and fossil-fuel heating in the winter, etc. The local topography and terrain and even autumn and winter meteorological conditions also contribute to the gathering of pollution. It is not conducive and also exists in the Yangtze River Delta and Sichuan Basin regions [56,57].

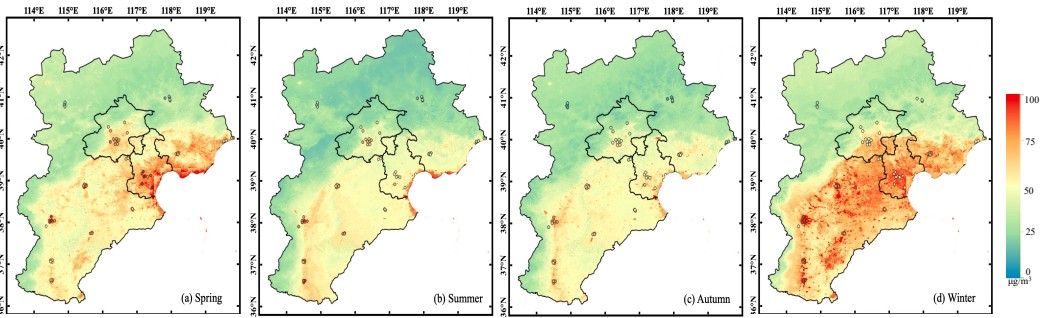

**Figure 8.** Spatial distribution of seasonal averaged PM$_{2.5}$ concentrations in 2017.

Comparing the spatial distribution values with the monitoring point values, the overall PM$_{2.5}$ spatial distribution accuracy is similar to that of the corresponding ground monitoring points. Among them, the estimated values of Beijing and Tianjin are relatively more accurate, which may be attributed to the relatively large number of air quality monitoring stations and more uniform spatial distribution in these two cities. On the contrary, the ground stations in Hebei province are sparsely distributed in each city and the number of stations is small, so the visual comparison is slightly underestimated. In addition, a significant overestimation of PM$_{2.5}$ occurs along the Bohai Sea coast, which may be due to the difference in aerosol types along the seashore, model training data obtained from ground monitoring sites that are in the land area, and meteorological conditions such as sea fog and sea wind. This also indicates that the data-driven model relies on the complete data set with high accuracy.

This study cross-validated the hourly PM$_{2.5}$ estimates with ground monitoring values for each season in 2017, and the relevant scatter plots are shown in Figure 9. It can be seen that although the highest RMSE (11.841 µg/m$^3$) and MAE (7.909 µg/m$^3$) indicators for PM$_{2.5}$ were found in summer, the R$^2$ was only 0.783. On the contrary, the lowest RMSE (24.937 µg/m$^3$) and MAE (13.611 µg/m$^3$) indicators were found in winter, but the R$^2$ reached 0.912. The scatter plot also shows the lowest PM$_{2.5}$ concentration values in summer (about 0–150 µg/m$^3$) and the highest PM$_{2.5}$ concentration values in winter (about 0–500 µg/m$^3$), and therefore, the estimated PM$_{2.5}$ in winter has difficulty reflecting the advantage in RMSE and MAE indexes, and the measurement of model performance should be compared comprehensively. In addition, the indicators are closer in spring (R$^2$ = 0.852, RMSE = 17.976 µg/m$^3$, MAE = 9.502 µg/m$^3$) and autumn (R$^2$ = 0.879, RMSE = 15.105 µg/m$^3$, MAE = 9.243 µg/m$^3$). The differences in model performance for different seasons of PM$_{2.5}$ may have some correlation with the sample size and seasonal natural conditions. For example, on the one hand, in BTH region, there is more snow and ice cover in winter and cloudy and rainy weather in summer. Those differences have a greater impact on the number of effective samples, thus the samples in winter and summer will be significantly less than that in spring. On the other hand, summer weather conditions are conducive to the rapid dissipation of pollution, making it difficult to form aggregated pollution. The autumn and winter seasons are closely related to the increase in pollution emissions, and the Beijing–Tianjin–Hebei region is more influenced by the northwest dust in spring, which also makes the nonlinear relationship between AOD and PM$_{2.5}$ more

complicated. It is difficult for the model to completely fit such complex relationships without multi-source data and the attention mechanism.

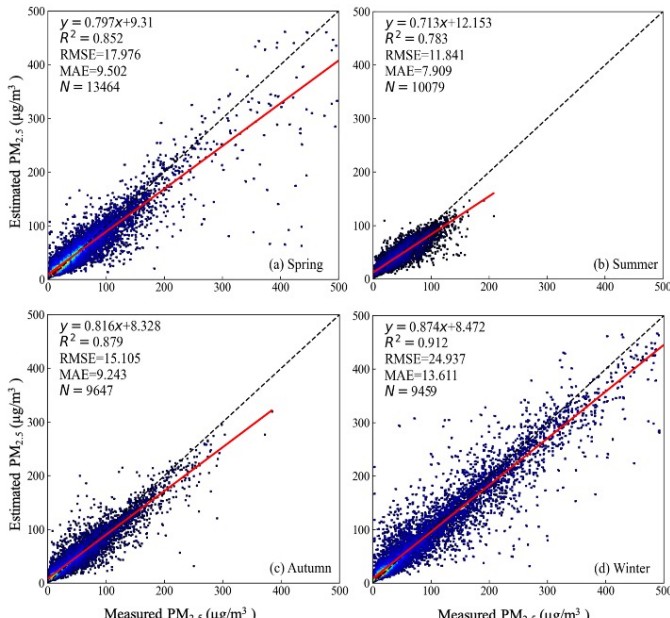

**Figure 9.** Density scatterplots of cross-validation correlation between estimated hourly $PM_{2.5}$ concentrations and ground measurements in 2017.

Table 4 presents statistics on the detailed model-estimated average $PM_{2.5}$ concentrations at different hours in 2017 compared with the corresponding ground-based measurements for the validation results. It can be seen that the proposed method in this study has a slightly different performance in each hour, but the overall main evaluation indexes do not differ much. The main reason for this may be due to the reduced number of training samples from optical remote sensing satellites at sunrise (8:00 vs. 9:00) and sunset (16:00 vs. 17:00).

In addition, there are significant daily variations in air pollution at different $PM_{2.5}$ pollution levels due to the difference in human activity intensity and natural conditions at each time period. The slope metric shows that the model has a certain degree of underestimation, which is a common problem in $PM_{2.5}$ estimation studies, and some scholars believe that this is due to the large uncertainty in the aerosol retrievals and the small data samples under high pollution conditions [8,22,56].

**Table 4.** Cross-validation of estimated and measured average $PM_{2.5}$ at different hours.

| Time | Samples | $R^2$ | RMSE | MAE | Slope | Estimated | Measured |
|------|---------|-------|------|-----|-------|-----------|----------|
| 08:00 | 3099 | 0.817 | 16.32 | 10.52 | 0.72 | $50.2 \pm 27.8$ | $49.3 \pm 34.7$ |
| 09:00 | 4739 | 0.820 | 17.13 | 11.24 | 0.76 | $52.3 \pm 32.9$ | $49.8 \pm 39.2$ |
| 10:00 | 6931 | 0.855 | 22.19 | 12.11 | 0.78 | $57.4 \pm 50.2$ | $55.6 \pm 59.4$ |
| 11:00 | 7244 | 0.881 | 20.61 | 11.81 | 0.83 | $55.1 \pm 51.7$ | $53.2 \pm 58.6$ |
| 12:00 | 7188 | 0.884 | 19.46 | 10.82 | 0.85 | $51.3 \pm 51.1$ | $49.8 \pm 56.6$ |
| 13:00 | 6953 | 0.902 | 18.01 | 9.87 | 0.86 | $50.0 \pm 51.7$ | $49.7 \pm 57.0$ |
| 14:00 | 6848 | 0.891 | 19.02 | 10.54 | 0.85 | $49.4 \pm 51.0$ | $50.6 \pm 56.9$ |
| 15:00 | 6550 | 0.903 | 18.84 | 11.01 | 0.84 | $49.1 \pm 52.2$ | $52.2 \pm 58.7$ |
| 16:00 | 4500 | 0.878 | 18.76 | 10.83 | 0.80 | $44.6 \pm 44.6$ | $49.3 \pm 51.2$ |
| 17:00 | 2814 | 0.745 | 16.49 | 10.68 | 0.69 | $36.3 \pm 22.3$ | $41.9 \pm 30.2$ |
| ALL | 56,866 | 0.873 | 18.60 | 11.92 | 0.83 | $50.7 \pm 48.1$ | $50.9 \pm 54.6$ |

Figure 10 illustrates the spatial and temporal distribution of hourly average PM$_{2.5}$ concentrations in 2017, where the point source distribution data are ground monitoring values. As can be observed in the figure, the spatial distribution of pollution across the study area is characterized by the same seasonal scale, with low PM$_{2.5}$ pollution in the northwest region and high pollution in the southeast. From the annual average spatial distribution of PM$_{2.5}$ at different hours, the trend of PM$_{2.5}$ pollution shows the characteristics of rising first and then falling from morning to evening. The pollution reaches a peak around 11:00 a.m., and PM$_{2.5}$ pollution drops to a relatively low level in the evening, which may be caused by the increase in pollution emission in the morning and the superposition effect of pollution gathering from the previous night, etc. This kind of PM$_{2.5}$ pollution spatial and temporal distribution characteristic is also similar to some existing studies [53], especially that the increase in pollution emissions caused by anthropogenic activities, industrial production, etc., during the corresponding time of autumn and winter seasons are more relevant.

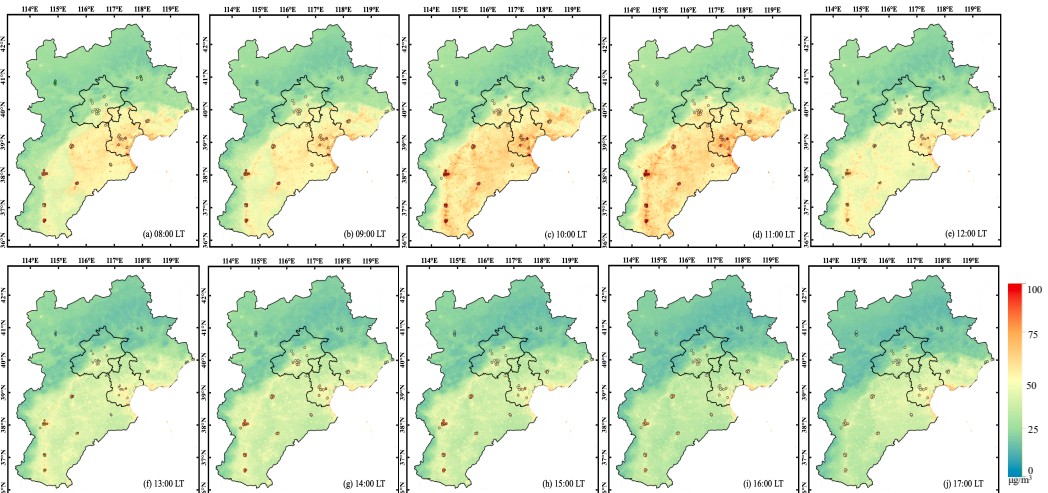

**Figure 10.** Spatial distribution of averaged PM$_{2.5}$ concentrations at different hours (08:00 to 17:00 local time) during 2017.

## 4. Conclusions

In this paper, we improved the estimation of PM$_{2.5}$ concentration across Beijing–Tianjing–Hebei regions using multi-source data by introducing the attention mechanism into random forest. The attention mechanism was used to co-weight the environmental similarity and the geographical distances, and the RF algorithm was applied to estimate hourly PM$_{2.5}$ of BTH based on the Himawari-8/AHI AOD products. The experiment results demonstrated that our approach can more efficiently characterize the influence of spatio-temporal information hidden in adjacent ground monitoring sites. The hourly PM$_{2.5}$ estimates are well correlated with ground measurements in BTH, with an R$^2$ of 0.887, an RMSE of 18.31 μg/m$^3$, and an MAE of 11.17 μg/m$^3$, indicating the good model performance. Furthermore, to simplify the model structure and improve the efficiency while maintaining accuracy, this paper thoroughly examines of the effectiveness of multi-source data in the estimation process, including the analysis of the effect of AOD data quality on the final estimation accuracy of the model and the ranking results of the feature importance of the random forest. Finally, this approach provides full-coverage PM$_{2.5}$ estimation in BTH by fusing satellite and model data.

In future work, we will pay attention to computation efficiency and model robustness, then expand the estimation area with reasonable spatial–temporal analysis and validation.

**Author Contributions:** Conceptualization, L.Z. and Z.L.; methodology, L.Z., J.G., and Y.X.; software, L.Z. and H.G.; validation, L.Z., Z.S. and Y.Z.; formal analysis, L.Z., Z.L., and J.G.; writing—original draft preparation, L.Z.; writing—review and editing, J.G. and Y.X.; visualization, L.Z., Y.Z., and H.G.; supervision, Z.L. All authors have read and agreed to the published version of the manuscript.

**Funding:** This work was supported by the Finance science and technology project of Hainan province (science and technology cooperation project, Key R&D plan of Hainan Province, ZDYF2020206), the National Natural Science Foundation of International Cooperative Research Project (Grant No. 42111530292), and the National Natural Science Foundation of China (Grant No. 42175147).

**Institutional Review Board Statement:** Not applicable.

**Informed Consent Statement:** Not applicable.

**Data Availability Statement:** The data presented in this study are available on request from the corresponding author. The data are not publicly available due to privacy.

**Acknowledgments:** Thanks to the agencies that provided the data in this study. The in situ PM$_{2.5}$ ground measurements are available from the China National Environmental Monitoring Center (http://www.cnemc.cn, accessed on 7 February 2024). The Himawari-8 satellite data were downloaded from the website (https://www.eorc.jaxa.jp/ptree/, accessed on 7 February 2024). MODIS data were obtained from LAADS DAAC (https://ladsweb.modaps.eosdis.nasa.gov, accessed on 7 February 2024), and ERA-5 reanalysis data were obtained from the European Centre for Medium-Range Weather Forecasts (http://apps.ecmwf.int/datasets/, accessed on 7 February 2024).

**Conflicts of Interest:** The authors declare no conflicts of interest.

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
