# Peer review of "Improving the Estimation of PM2.5 Concentration in the North China Area by Introducing an Attention Mechanism into Random Forest"

_atmosphere, doi:10.3390/atmos15030384_

Round 1
Reviewer 1 Report
Comments and Suggestions for Authors
The proposed paper Improving the Estimation of PM2.5 Concentration in North China Area by Introducing Attention Mechanism into Random Forest proposed an interesting study dealing with estimation of PM2.5 in limited area such as the Beijing area in China. The paper is well motivated and the conclusions are clearly described. I also highly appreciate the comparison with ground measurements where good agreement is achieved. In sum, this is well written paper and I recommend it for publication.
Author Response
Dear Reviewer,
We are very grateful for your time and effort to review our manuscript.
Thank you for your recognition of our research.
Reviewer 2 Report
Comments and Suggestions for Authors
1- Why did you use 90m DEM data? there are higher resolution data (e.g., 30m)
2- What is the time interval used in this study?
3- In Fig. 4f, hte texts are not clear to read.
4- How did you set the parameters in each algorithm?
5- In Fig.7, what is the difference betwenn two lines (dash and straight line)
6- What is the line which is under the x-axis of AERONET AOD in Fig. 7 and 9?
7- Why the number of samples in different time in Table. 4 is too different?
8- Add color-bar for Fig. 2.
9- Is there any reference in 2023 in this scope that you haven't mentioned or used for literature review?
Author Response
Dear Reviewer,
We are very grateful for your time and effort to review our manuscript and provide constructive comments and suggestions, which undoubtedly helped to improve the quality of the manuscript. In the revision, we made the changes, as you suggested.

Reviewer 3 Report
Comments and Suggestions for Authors
The research is well-designed and well-executed. However, the presentation of the research needs improvement especially the methods section where the introduction and discussion parts details are added. They should be moved to their respective sections and the methods section should be simple and precise.

The quality of the English language is appropriate.
Author Response

(The authors gave the same response as above.)

Round 2
Reviewer 2 Report
Comments and Suggestions for Authors
This paper can be published in the present form.
Reviewer 3 Report
Comments and Suggestions for Authors
The modified version has improved presentation of the research